

**PeerJ Hubs**
Published on behalf of

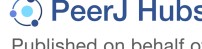


# Profiling microplastic fibers in the intertidal sentinel mussel *Brachidontes rodriguezii* from the coast of Buenos Aires, Argentina

Lucina Olivia Migliarini[1], Sonia Maribel Landro[1], Freddy Martinez-Espinoza[2], Daniel Horacio Murgida[2] and Florencia Arrighetti[1]

[1] Laboratorio de Ecosistemas Marinos, Museo Argentino de Ciencias Naturales "Bernardino Rivadavia" -CONICET, Buenos Aires, Argentina
[2] Departamento de Química Inorgánica, Analítica y Química Física and INQUIMAE, Facultad de Ciencias Exactas y Naturales, Universidad de Buenos Aires-CONICET, Buenos Aires, Argentina

## ABSTRACT

Mussels can accumulate microplastics (MPs) present in seawater and are one of the species most affected by MP pollution. This study is the first to evaluate the abundance of MPs in the small mussel *Brachidontes rodriguezii* at four stations (S1, S2, S3, and S4) with different levels of human activities along the intertidal area of the most popular resort city of Argentina (Mar del Plata, Buenos Aires). Microplastics, primarily microfibers, were detected in 97.5% of the analyzed mussels by visual identification. The abundance of MPs varied significantly among the stations, with the highest levels observed in mussels from S4, corresponding to the low-urbanized area. This finding seems to suggest that factors other than urban pollution, such as agricultural activities and nearby streams, may contribute to MP contamination. The study also found a relation between MPs abundance and the mussels' condition index, suggesting that high levels of MPs may negatively impact the health of these organisms. Identification suggested that all found microfibers were plastic, with approximately 10% of the analyzed microfibers revealing the presence of polymers such as polyester, polychloroprene, polyacrylonitrile, and polyethylene terephthalate. For several microfibers, only the pigments but not the substrate could be identified, and about half of the microfibers were Raman inactive, thus limiting definitive identification. These findings highlight the widespread MPs contamination in marine environments and the use of mussels as bioindicators of MP pollution. Future research should focus on identifying the sources of MPs, assessing their potential ecological impacts, and developing effective strategies for mitigating MP pollution.

# INTRODUCTION

Microplastics (MPs), plastic particles <5 mm in size, are a major global problem due to their abundance in the marine environment (*GESAMP, 2015*). Because of their widespread use and persistence, MPs are present in all marine environments, including sediments,

Corresponding author
Florencia Arrighetti,
florarrighetti@gmail.com

seawater, and biota (*Kamyab et al., 2018*). Most MPs enter the marine environment through terrestrial sources such as water runoff, coastal human activities, and mainly, through wastewater treatment plants (*Sun et al., 2019*). These MPs range in size from small to large and occur in various colors (transparent, blue, black, red, *etc.*). They also display diverse morphotypes, including fibers, pellets, fragments, and films. The toxicity of plastic particles depends on both the quantity and physical characteristics of ingested items, as well as their chemical properties (*Leistenschneider et al., 2023*). MPs comprise different polymer compositions such as polystyrene (PS), polypropylene (PP), nylon, polyethylene terephthalate (PET), or polyvinyl chloride (PVC). Chemical characterization can be performed using various spectroscopy techniques, including micro-Fourier transform infrared spectroscopy ($\mu$-FTIR), attenuated total reflection Fourier transform infrared spectroscopy (ATR-FTIR), or micro-Raman spectroscopy ($\mu$-Raman) (*Bessa et al., 2019*).

Fibers, also called microfibers, are the most predominant morphotype observed in aquatic ecosystems and can cause several adverse effects on the biota (*Kwak II et al., 2022*). Additionally, their prolonged egestion times in marine organisms rise heightened concern, as this may result in increased exposure and associated risks (*Cole et al., 2016*). MPs ingestion causes not only physical but also physiological damage on marine organisms (*Derraik, 2002*; *Andrady, 2015*; *Lusher, 2015*; *Crawford & Quinn, 2017*). Due to their filter-feeding mechanism, bivalves accumulate MPs from the water column (*Li et al., 2016*; *Reguera, Viñas & Gago, 2019*). Therefore, the detection of MPs in their soft tissues serves as an indicator of the concentration and bioavailability of this pollutant in the environment (*Beyer et al., 2017*). Furthermore, the accumulation of MPs in bivalves is of human health concern because they can enter the human food supply (*Li et al., 2016*).

However, establishing a direct relationship between MP accumulation in organisms and environmental concentrations presents significant challenges compared to conventional pollutants, primarily due to the variability in size and polymer characteristics of MPs (*Beyer et al., 2017*; *Qu et al., 2018*; *López-Monroy & Fermín, 2019*). To overcome these difficulties, studies must follow standardized methodological protocols and should prioritize methodologies that include advanced polymer identification (*Bessa et al., 2019*). This enables comparisons between studies, a better understanding of the potential consequences of the MPs found, and ultimately, proper mitigating and potential risk management proposals.

Despite the aforementioned concerns, studies on the presence of MPs in marine mussels are still scarce in Argentina (*Pérez et al., 2020*; *Ríos, Hernández-Moresino & Galván, 2020*; *Truchet et al., 2021*). The intertidal small mussel *Brachidontes rodriguezii* (d'Orbigny, 1846) is one of the dominant organisms on the rocky shores of Argentina (*Adami, Tablado & López-Gappa, 2004*; *Arribas et al., 2015*) and plays a critical ecological role because it forms "mussel matrices" that are assemblies of other benthic species that include polychaetes, algae, and crustaceans among other taxa (*Elías et al., 2006*). In addition, *B. rodriguezii* has been used as a biological indicator of pollution in Buenos Aires (Argentina) coastal areas (*Arias et al., 2009*; *Oliva et al., 2015*; *Oliva et al., 2017*; *Laitano et al., 2016*; *Buzzi et al., 2017*; *Quintas et al., 2017*; *Buzzi & Marcovecchio, 2018*; *Arrighetti et al., 2019*). However, there is only one study regarding MP concentrations in this mussel species (*Truchet et al., 2021*).

The city of Mar del Plata, in Buenos Aires, represents the major seaside resort area in Argentina and, although it has a permanent population of 618,989 inhabitants (*National Institute of Statistics and Census of Argentina (INDEC), 2022*), it receives more than eight million tourists during the summer season (*Miglioranza et al., 2021*; *Municipality of General Pueyrredón (MGP), 2023*). The port of Mar del Plata serves not only as one of Argentina's most important commercial and fishing ports but also as a tourist attraction in the area. Pollution in the area has been assessed in sediments (*Albano et al., 2013*; *Laitano et al., 2015*) and different invertebrate species (*Rivero, Elías & Vallarino, 2005*; *Laitano & Fernández-Gimenez, 2016*; *Arrighetti et al., 2019*; *Landro, Teso & Arrighetti, 2021*). A water treatment plant (WTP) located in the neighborhood of Camet treats the household effluents of Mar del Plata City. Before releasing the treated water into the sea through a submarine outfall, this facility filters and treats the sediments, sands, fats, and oils from the sewage effluent (*Sanitary Works Department of Argentina (OSSE), 2020*; *Miglioranza et al., 2021*). Because of the increased population during the tourist seasons, there is a significant seasonality in the discharge rate of the WTP (*Scagliola & Furchi, 2006*). This study aims to report, for the first time, the abundance and chemical characterization of MPs in the small mussel *B. rodriguezii* in several areas of Mar del Plata city and to evaluate the influence of the different anthropogenic pressures on the abundance of MPs. We hypothesized that mussels closer to the sewage treatment plant outfall area present a greater abundance of MPs than the other study locations since these particles could pass through the sewage treatment screens and be released into the aquatic environment.

## MATERIALS & METHODS

### Study sites and mussel collection

Based on the different anthropogenic activities that impact the area, four sampling sites were selected: Camet (S1), Port (S2), Lighthouse (S3), and Brusquitas (S4) (Fig. 1). S1 is located 21 km north of Mar del Plata city in a moderately urbanized area next to a sewage treatment plant, with an underwater outfall that discharges wastewater four km from the coast. S2 is located in one of the most important fishing ports of the country, which is in the main tourist and commercial area of Mar del Plata and receives great industrial and urban pressures. S3 is a moderately urbanized site located near a lighthouse on the south of Mar del Plata, while S4 is located near the Brusquitas stream, which has a permanent water regime and is located in a low urbanized area in the southern limit of Mar del Plata city (Table 1). The sites with high-intensive anthropic pressure activities, *i.e.,* S1 and S2, were considered highly contaminated, while the sites with less anthropic pressure, *i.e.,* S3 and S4, were considered slightly contaminated sites (*Arrighetti et al., 2019*).

*B. rodriguezii* adult mussels ($n = 10$ per site) were collected randomly from the intertidal rocky shore during low tide in August 2018. Samples were collected manually using stainless steel tools, placed in sealed plastic bags, and then transported to the laboratory, where they were stored at $-20\,^{\circ}\mathrm{C}$.

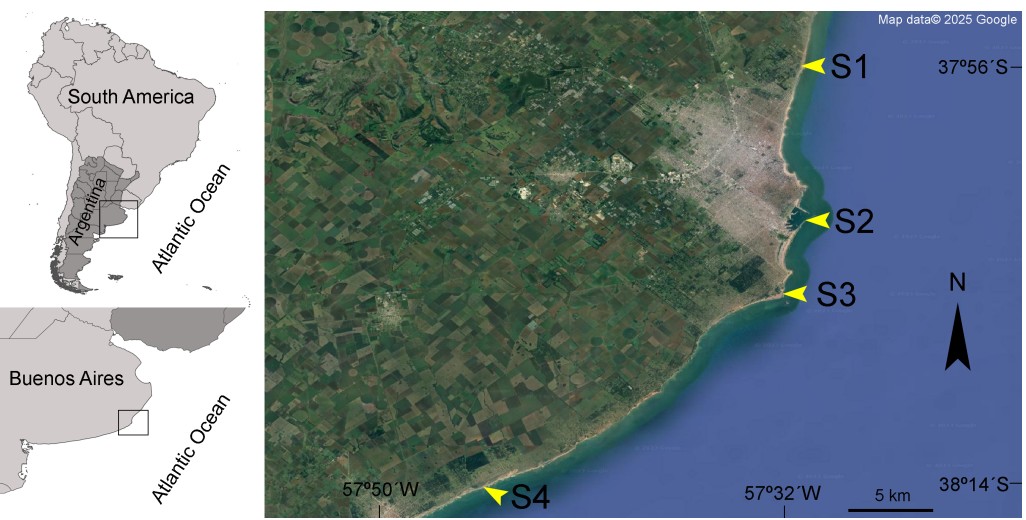

**Figure 1 Studied area.** Geographical location of the four sampling sites along the Mar del Plata city coast in Buenos Aires, Argentina. Camet (S1), Port (S2), Lighthouse (S3), and Brusquitas (S4). Map data ©Google.

**Table 1 Details of the sampling sites, morphometric characteristics, and condition index of mussels *Brachidontes rodriguezii* collected in each study site.**

| Site | Location | Geographical location | Mean shell length (mm) | Mean soft tissue weight (g) | Condition index |
|------|----------|----------------------|------------------------|------------------------------|-----------------|
| S1 | sewage treatment plant | 37°54′21.07″S 57°31′21.22″W | 21.41 ± 1.89[a] | 0.29 ± 0.06[a] | 34.51 ± 4.28[a] |
| S2 | port | 38°01′44.08″S 57°31′52.55″W | 23.13 ± 2.53[a] | 0.28 ± 0.06[a] | 28.74 ± 2.59[bc] |
| S3 | moderately urbanized area | 38°05′42.08″S 57°32′31.69″W | 21.94 ± 1.24[a] | 0.27 ± 0.06[a] | 31.48 ± 2.50[ab] |
| S4 | low urbanized area | 38°14′43.47″S 57°46′41.00″W | 17.36 ± 1.15[b] | 0.10 ± 0.03[b] | 24.36 ± 5.59[c] |

**Notes.**
Values are expressed as mean ± SD. Different letters indicate statistically significant differences among sites ($p < 0.05$).

## Microplastic isolation

The analysis of MPs in mussels was performed according to standardized protocols (*Bråte et al., 2018a*; *Reguera, Viñas & Gago, 2019*) with some modifications. Mussels were defrosted, their byssus was removed, and the individuals were thoroughly rinsed with double-filtered water. The total shell length was measured with a caliper and the total weight was registered using a precision electronic scale. The soft tissue was carefully dissected with a scalpel and weighed to assess a mussel's condition index (CI), which was calculated using the following equation (*Arrighetti et al., 2019*):

$$CI = ST/TW; \tag{1}$$

where ST is the soft tissue wet weight (g) and TW is the total mussel wet weight (g). Then, the soft tissues from each individual mussel were added to separate sealed glass beakers

containing 20 mL of 10% KOH per gram of mussel tissue. Samples were digested in an incubator at 60 °C and agitated at 140 rpm for 8 h. After digestion, each sample was filtered through a gridded cellulose nitrate membrane filter (Microclar®, pore size 1.2 μm diameter) using a vacuum pump. The filters were stored in sealed Petri dishes and dried at room temperature. Visual inspection of the filters was made using a stereomicroscope (Leica MZ95) and images of the potential MP particles were taken with a camera (Leica IC80 HD). Potential MPs were selected based on morphological criteria that included small size, absence of cellular structure, homogeneous color and structure (*Crawford & Quinn, 2017*). Finally, the potential MPs were counted and classified according to their shape and color.

Digestion efficiencies (%*De*) were calculated as the percentage of tissue remaining after digestion according to the following equation:

$$\%De = 100 - ((DW_{\text{fad}} - DW_{\text{f}})/T_{\text{w}}) \times 100; \tag{2}$$

where $DW_{\text{fad}}$ and $DW_{\text{f}}$ correspond to the dry weights of the filter covered by organic matter and debris after digestion, and the "clean" filter before filtration, respectively. $T_{\text{w}}$ corresponds to the mean weight of tissue subjected to digestion (*Dehaut et al., 2016*).

MP abundance was expressed as the number of MPs items/individual and the number of MPs items/g soft tissue wet weight (w/w).

## Chemical characterization of microplastics by Raman spectroscopy

To identify the chemical composition of the potential MPs observed under the stereomicroscope, a random subsample of 15% of the specimens was analyzed using a confocal Raman microscope (Horiba LabRAM HR Evolution) equipped with a front-illuminated CCD detector, a 600 l/mm grating and a long working distance 50x objective. Filters were placed on the XY motorized microscope stage (MW Tango) using a homemade holder. An attempt was made for all specimens to measure Raman spectra using a 532 nm laser (1 mW in the sample). In cases where no signal was observed, a second attempt was made with a 405 nm laser (150 μW in the sample). Particles were spotted at up to three points when no or poor signals were detected. Spectra reported here are the averages of two consecutive 5-second accumulations. After background subtraction using LabSpec 6 software, spectra were compared with open-access reference libraries (*Munno et al., 2020*; *Miller et al., 2022*) and/or specific literature for the identification, as indicated in the Results.

## Contamination precaution and quality control measures

As airborne contamination is a frequent problem in MPs studies (*Foekema et al., 2013*), strict quality controls were conducted during the collection, processing, and analysis to avoid any possible contamination of the samples. Laboratory precautions included clean laboratory conditions in an enclosed room, the use of cotton laboratory coats, and nitrile gloves. All the laboratory materials were washed with double-filtered water through glass microfiber filters and covered with aluminum foil. During the digestion procedure, negative controls were carried out with 50 mL of 10% KOH and without tissue samples, and during

the visual inspection, the filters were covered with glass lids. Additionally, an air control filter was left exposed during the procedures to account for airborne contamination.

## Statistical analysis

The results were reported as mean ± standard deviation (SD). Statistical analyses were performed using the R software v. 4.4.1 (*R Core Team, 2021*), with a significance level of $p < 0.05$. The normality of data was tested with the Shapiro–Wilks test and the homogeneity of variances with the Bartlett test. When the assumptions were met, a one-way analysis of variance (ANOVA) was performed to evaluate differences among study sites. Otherwise, a non-parametric Kruskal–Wallis test was performed. If significant differences were observed, a Tukey's multi-comparison test or pairwise comparisons with Bonferroni correction were used (after the ANOVA and Kruskal–Wallis tests, respectively).

## RESULTS

The mean shell length of the total analyzed mussels was $20.96 \pm 2.79$ mm and the mean soft tissue wet weight was $0.23 \pm 0.10$ g. The morphometric data of the collected mussels from each site are presented in Table 1. The mean shell length and the mean soft tissue wet weight of mussels collected in S4 were statistically lower than in the rest of the sites (ANOVA, $p < 0.05$, $F = 19.60$ and $F = 27.15$, respectively). Statistically significant differences were observed in the condition index (CI) among sites, with the highest value corresponding to mussels from S1 and the lowest to mussels from S4 (ANOVA: $F = 11.89$, $p < 0.05$) (Table 1).

The airborne control and negative control filters presented zero to two fibers. As the mean number of fibers in these controls did not exceed one, we could conclude that airborne contamination was effectively minimized. Therefore, these control values were not subtracted from the raw data (*Ding et al., 2021*). The average digestion efficiency was 99%. All the initially isolated particles (239 items) were identified as fibers—hereinafter called microfibers. Microfibers were found in 39 of the 40 mussels analyzed, with a frequency of occurrence of 97.50%. The abundance of microfibers per individual from all sites varied from 0 to 15 with a mean of $6.13 \pm 3.75$ items/individual. The abundance of microfibers per weight from all sites varied from 0 to 260 with a mean abundance of $39.77 \pm 49.62$ items/g ww. Regarding the color of the microfibers observed in all sites, the most frequent colors were transparent, followed by blue, red, black, and yellow (Fig. 2).

The highest microfiber abundance per individual was observed in mussels from S2 ($8.40 \pm 4.12$ items/individual) and S4 ($8.30 \pm 3.06$ items/individual). Both sites showed significantly higher values compared to S1 ($4.70 \pm 3.56$ items/individual) and S3 ($4.50 \pm 2.64$ items/individual) (ANOVA: $F = 7.20$, $p < 0.05$) (Fig. 3A). Additionally, the microfibers abundance per weight observed in mussels from S4 ($98.70 \pm 68.26$ items/g ww) was significantly higher than for the rest of the sites, followed by mussels from S2 ($30.14 \pm 14.41$ items/g ww) (Kruskal–Wallis: $H = 27.15$, $p < 0.05$) (Fig. 3B). A total of 41 microfiber were analyzed using Raman spectroscopy. Of these, 4 (9.75%) were univocally identified as MPs. One of these microfibers presented bands at 615 cm$^{-1}$, 842 cm$^{-1}$, 1,272 cm$^{-1}$, 1,600 cm$^{-1}$, 1,711 cm$^{-1}$, and 3,070 cm$^{-1}$ (Fig. 4A), which allows

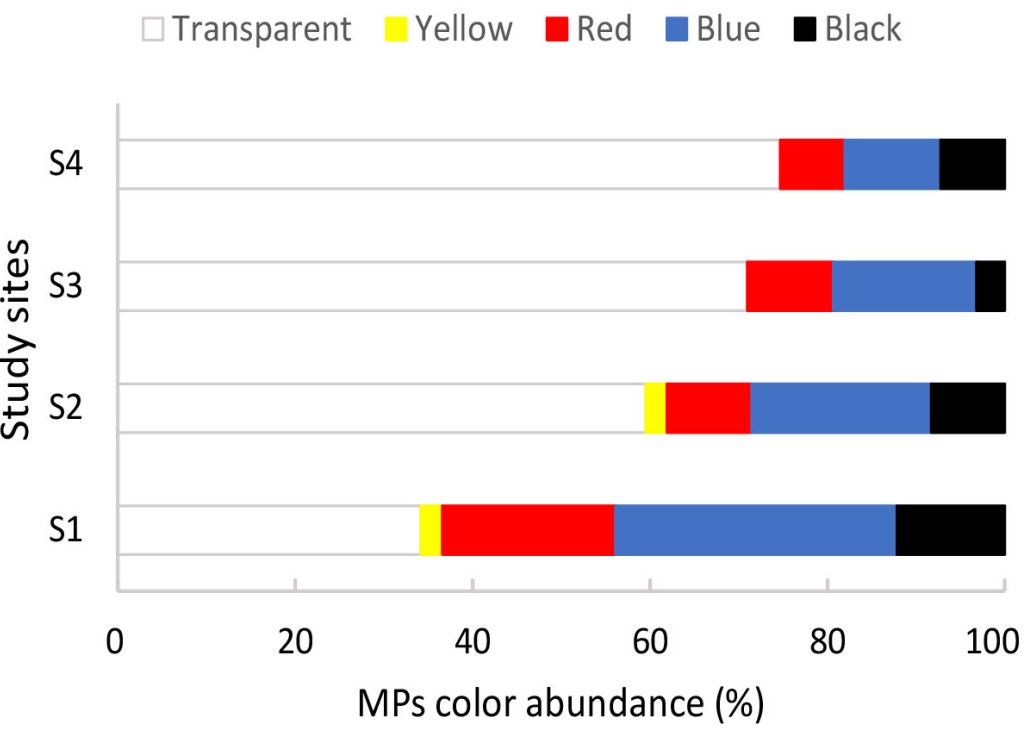

**Figure 2** Percentage of different colors of microplastics observed in mussels *Brachidontes rodriguezii* from the four study sites.

it to be assigned as polyester. The remaining bands, marked with an asterisk in the figure, correspond to pigments and additives. The spectrum of this microfiber is identical to that of a commercial orange polyester yarn. A second fiber presented bands at 466 cm$^{-1}$, 959 cm$^{-1}$, 999 cm$^{-1}$, 1,096 cm$^{-1}$, 1,178 cm$^{-1}$, 1,451 cm$^{-1}$, and 2,844 cm$^{-1}$ (Fig. 4B), corresponding to polychloroprene (*Gantmacher & Medvedev, 1943*; *Petcavich & Coleman, 1980*; *Arjunan, Subramanian & Mohan, 2003*; *Sathasivam, Haris & Mohan, 2003*; *Gupta et al., 2011*). The third particle was assigned as polyacrylonitrile due to its Raman signals at 1,079 cm$^{-1}$, 1,451 cm$^{-1}$, 2,241 cm$^{-1}$, 2,869 cm$^{-1}$, 2,915 cm$^{-1}$, 2,947 cm$^{-1}$, and 2,980 cm$^{-1}$ (Fig. 4C) (*Wang et al., 1996*; *Nava, Frezzotti & Leoni, 2021*; *Puchowicz & Cieslak, 2022*). The fourth microfiber was assigned as polyethylene terephthalate based on the characteristic Raman bands at 270 cm$^{-1}$, 620 cm$^{-1}$, 847 cm$^{-1}$, 1,084 cm$^{-1}$, 1,282 cm$^{-1}$, 1,406 cm$^{-1}$, 1,604 cm$^{-1}$, 1,715 cm$^{-1}$, 2,959 cm$^{-1}$, 2,996 cm$^{-1}$, and 3,075 cm$^{-1}$ (Fig. 4D) (*Munno et al., 2020*; *Nava, Frezzotti & Leoni, 2021*; *Miller et al., 2022*). A second group of 10 microfibers (24.38%) showed Raman signals that did not match with any polymer from the databases and literature consulted. Within this group, the spectra shown in Figs. 5A-5C correspond to the pigments phthalo blue 23050 (one fiber), indigo 36000 (three fibers), and Perinone PO43 CI71105 (one fiber), respectively (*Caggiani, Cosentino & Mangone, 2016*; *Infrared and Raman Users Group (IRUG), 2024*). However, the substrates colored by these pigments could not be identified. Three additional fibers presented Raman spectra consistent with either carbon microfibers or an unidentified microfiber colored with black bone 47100
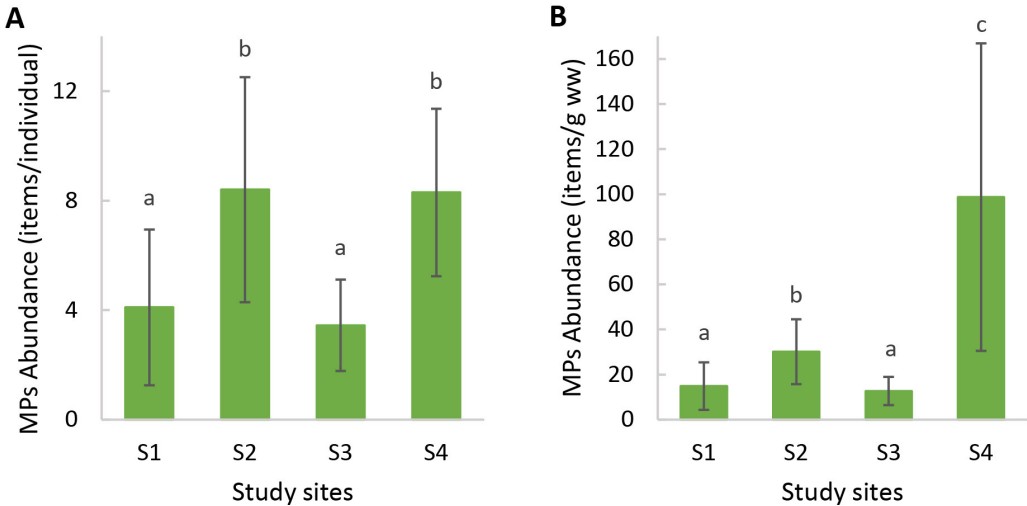

**Figure 3 Microplastic abundance (mean ± SD) in mussels *Brachidontes rodriguezii* from the study sites.** (A) Microplastic abundance per individual. (B) Microplastic abundance per gram of soft tissue wet weight. Different letters indicate statistically significant differences among sites ($p < 0.05$).

pigment or similar (Fig. 5D) (*Caggiani, Cosentino & Mangone, 2016*; *Infrared and Raman Users Group (IRUG), 2024*). Two fibers were identified as cotton (Fig. 5E) (*Munno et al., 2020*; *Puchowicz & Cieslak, 2022*). For the remaining microfibers, 21 (51.22%) showed no Raman activity with any of the available lasers, and 6 (14.63%) presented unassignable spectra.

# DISCUSSION

The present study reports, for the first time, the presence of MPs in the small mussel *B. rodriguezii* in the rocky intertidal area of Mar del Plata, Argentina. This species was chosen because it has been shown that it is a good bioindicator of different contaminants in the coastal environment (*Laitano & Fernández-Gimenez, 2016*; *Laitano et al., 2016*; *Buzzi & Marcovecchio, 2018*; *Arrighetti et al., 2019*; *Ojeda et al., 2021*). However, only one study has analyzed the presence of plastic polymers in this species (*Truchet et al., 2021*).

All the potential MPs observed in the mussels from our study were microfibers. Microfibers are part of several products in our daily life (*e.g.*, textiles, furniture, ropes), and are usually present in every marine habitat worldwide (*Gago et al., 2018*). Our results are in agreement with previous studies reporting fibers as the dominant morphology type found in bivalves (*Bråte et al., 2018b*; *Hermabessiere et al., 2019*; *Joshy, Krupesha Sharma & Mini, 2022*; *Klein et al., 2022*; *Khanjani, Sharifinia & Mohammadi, 2023*). In addition, microfibers are the most abundant type of MP particles found in the bivalves studied in Argentina (*Fernández Severini et al., 2019*; *Pazos, Spaccesi & Gómez, 2020*; *Pérez et al., 2020*; *Ríos, Hernández-Moresino & Galván, 2020*; *Truchet et al., 2021*). The predominance of microfibers may be attributed to their smaller size, which facilitates the ingestion and retention in mussels compared to larger MPs (*Van Cauwenberghe & Janssen, 2014*; *Zhang*

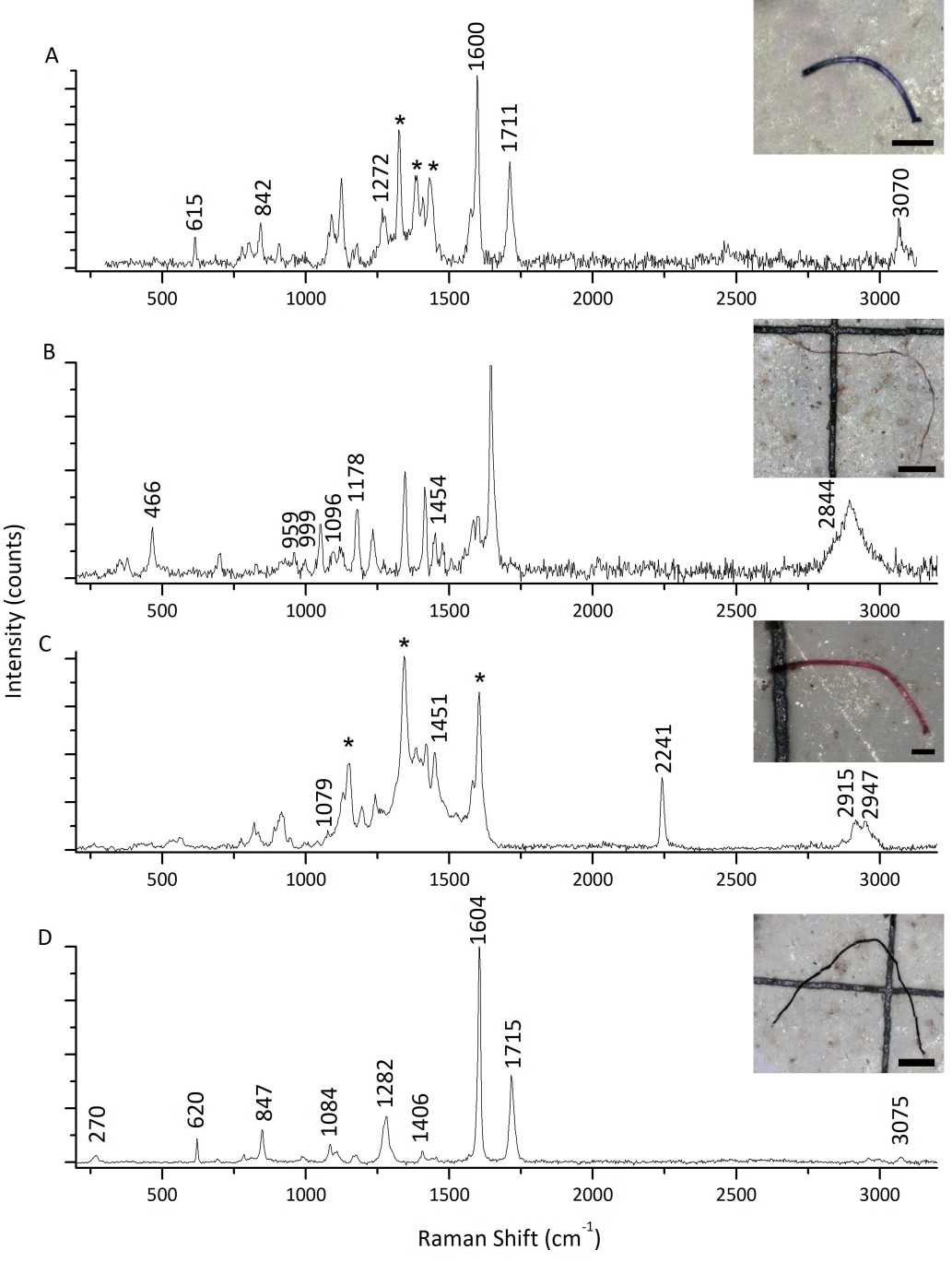

**Figure 4** **Raman spectra of four representative microfibers found in mussels *Brachidontes rodriguezii* from the four study sites.** (A) Polyester. (B) Polychloroprene. (C) Polyacrylonitrile. (D) Polyethylene terephthalate. Raman bands assigned to the MPs are indicated by their positions. Asterisks indicate spectral features that do not correspond to the main polymer, and most likely arise from pigments and/or additives (*Angelin et al., 2021*). All spectra shown here were acquired with 405 nm excitation. Scale bars: (A, C): 100 μm; (B, D): 400 μm.

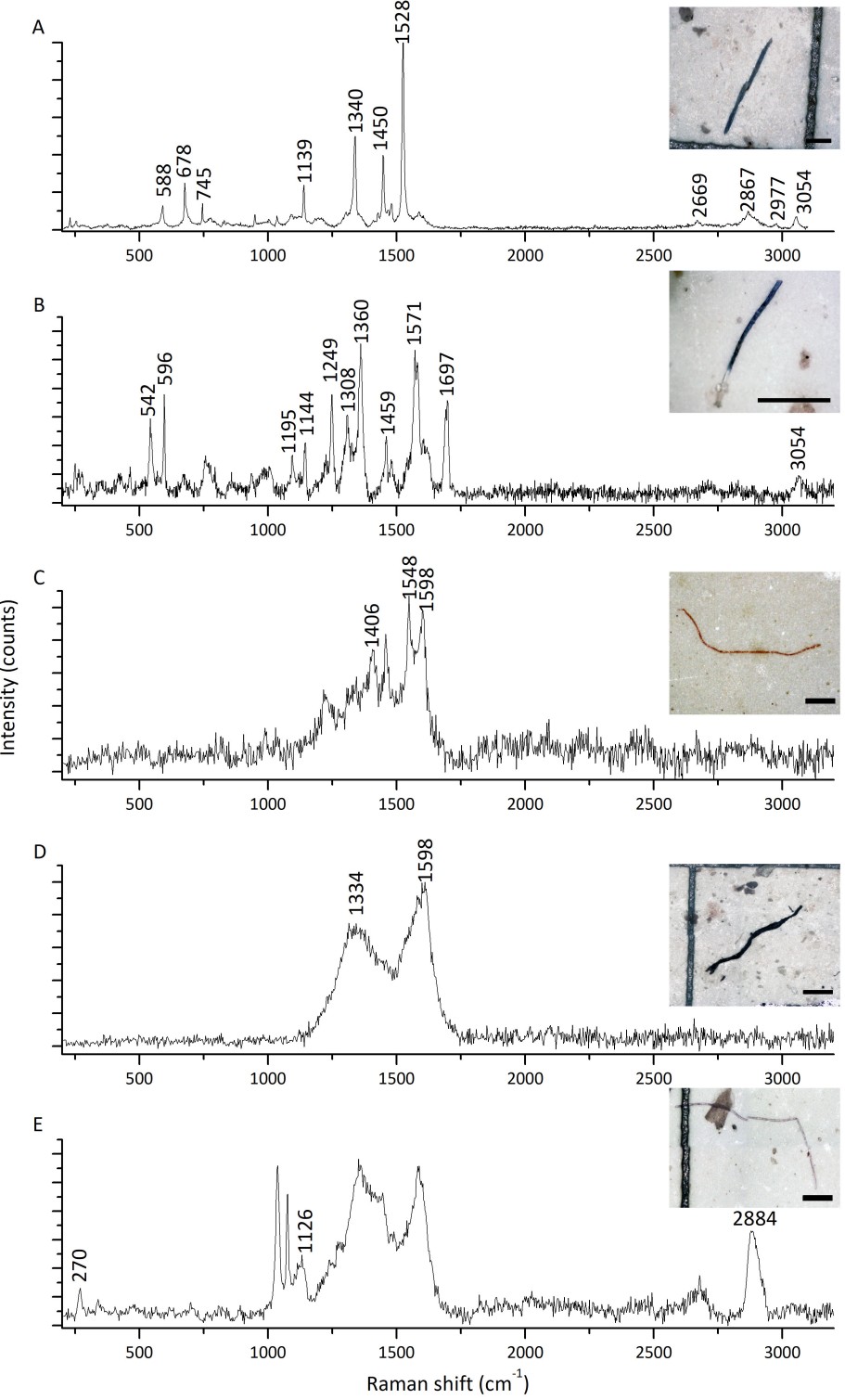

**Figure 5 Raman spectra of five representative microfibers found in mussels *Brachidontes rodriguezii* from the four study sites.** (A) Blue pigment (Phthalo Blue 23050). (B) Indigo pigment (Indigo 36000). (C) Orange pigment (Perinone: PO43 CI71105). (D) Consistent with either black pigment (Bone Black 47100) or carbon microfiber. (E) Cotton. Spectra A and B were recorded with 532 nm excitation, whereas C–D were acquired at 405 nm. Scale bars: (A, B, C): 100 μm; (D, E): 400 μm.

*et al., 2020*). According to *Browne et al. (2008)*, larger particles are easily removed from the digestive tract during the gut depuration period, while synthetic fibers take longer to be egested. This differential uptake and retention of microfibers could explain the variations in microplastic concentrations observed among mussels from different locations: sites with higher MP concentrations may present a greater abundance of smaller-sized MPs. Ultimately, this would also be determined by the relationship between the animals' feeding mechanisms, the behavior of the particles in the water column, and fibers entanglement tendency within the feeding structures (*Shimeta & Jumars, 1991*; *Murray & Cowie, 2011*; *Kolandhasamy et al., 2018*).

Our findings revealed that microfibers were present in almost all mussel samples from the four locations with different anthropogenic pressures (frequency of occurrence 97.50%). Furthermore, the mean total abundance of microfibers (6.13 ± 3.75 items/individual) in the present study was higher than in previous research on bivalves worldwide (for a summary see *Li et al., 2019*). For instance, mussels *Mytilus galloprovincialis* collected in the Mediterranean Sea showed a lower abundance of MPs (1.9 ± 0.2 items/individual) and a frequency of occurrence of 46.3% (*Digka et al., 2018*). Conversely, in mussels *Mytilus chilensis* from the coast of the Beagle Channel (Ushuaia, Argentina), *Pérez et al. (2020)* found a higher abundance (8.6 ± 3.53 items/individual) and an occurrence of 100%. *Truchet et al. (2021)* studied specimens of *Brachidontes rodriguezii* from remote beaches in the south of Buenos Aires Province, and they also found that microfibers were the most abundant type of MPs in these mussels (90%). However, the mean abundance they reported (0.17 ± 0.07 items/g ww) is considerably lower than the mean abundance from our study (39.77 ± 49.62 items/g ww). A direct comparison between the abundance of our research and the work by *Truchet et al. (2021)* is not possible due to methodological discrepancies. In the present study, we report MPs abundance based on the weight of mussels' soft tissue, excluding the valve, whereas *Truchet et al. (2021)* expressed their results based on the whole mussel weight (including the valve). Moreover, they used pooled samples, while our study analyzed MPs in individual mussels. These differences highlight the importance of following standardized protocols in MPs research. Implementing a standard approach, wich includes consistent sampling processing and reporting data in suitable units (studies should report both items/individual and items/animal wet weight), will improve the comparability of results on the spatial variations of MP contamination in mussels. Despite the limited direct comparisons within Argentine studies, the observed variations in MP contamination across different sites underscore the spatial heterogeneity of this pollutant, even among the same species. Such differences may be attributed to anthropogenic pressures prevalent in urbanized areas, including sewage discharge from treatment plants and activities associated with fishing ports, as evident in our study.

While mussels collected in S4 and S2 exhibited higher MP abundances, the levels found in mussels from S1 and S3 were also high compared to reports from other studies (*Li et al., 2019*). Contrary to our hypothesis, mussels collected from the low urbanized area (S4, Brusquitas) exhibited the highest abundance of microfibers per mussel wet weight. While urbanized areas typically present higher MPs abundance (*Wang et al., 2017*; *Long et al., 2019*; *Wang, Zhao & Xing, 2021*), as evidenced by the mussels from the harbor area (S2),

we are alarmed by the magnitude of MP contamination in S4. This high level of MPs turns S4 into a potential MPs hotspot. Several factors could be causing this unexpected result. For instance, the Brusquitas stream near S4 runs through areas with extensive agricultural activities (*Hines et al., 2023*). Agricultural development often involves plastic greenhouses and shading systems, which can contribute to plastic waste generation (*Arias et al., 2020*). As the stream flows through the area, MPs might be collected from these agricultural sources. Furthermore, the stream crosses roads, rural paths, and public recreation areas, which could introduce additional plastic debris into the waterway. *Ronda et al. (2023)* studied the effects of tourism and leisure activities on MPs abundance on different sandy beaches of Argentina and observed the highest abundance of MPs on a beach with very little influence from tourism. The authors point out that sources of river discharges linked to littoral drift may also be an important factor for MPs input onto beaches and that agricultural land use and irrigation play a significant role in MP distribution (*Ronda et al., 2023*). To understand the unexpected high abundance of microfibers found in the low urbanized area of this study, further research involving water and sediment samples is warranted. This will help determine whether the Brusquitas stream acts as a point source of MP pollution in the area.

Mussels collected in S4, which exhibited the lowest condition index (CI), were also the smallest and lightest of all sites. *Catarino et al. (2018)* found a negative significant correlation between the number of ingested particles and the mussels' body weight. Specifically, smaller individuals, which are characterized by higher pumping rates due to their metabolic demands, tend to ingest a greater volume of water, and consequently, more MPs. For example, studies have shown that juvenile bivalves exhibit increased filtration rates relative to their body size, leading to higher particle encounter rates (*Riisgård, 2001*). Furthermore, smaller mussels might have a lower egestion efficiency of MPs, potentially retaining MPs for longer periods which could be attributed to a less developed digestive system or a slower gut passage rate compared to larger individuals. These factors suggest that smaller mussels are more vulnerable to MP accumulation, warranting further investigation into the physiological mechanisms governing uptake and egestion in different size classes of animals.

The CI is usually an indicator of mussels' health status (*Arrighetti et al., 2019*), raising questions about the potential relationship between microfiber abundance and mussels' condition. The high microfiber abundance observed in S4 may cause stress and impair the mussels' health. Future studies are required to understand how MP accumulation varies with mussels' body size and the CI. In addition, mussels undergo physiological and biochemical changes throughout their life cycle and across seasons (*Bråte et al., 2018b*; *Sendra et al., 2021*), which can influence their sensitivity to MPs. These variations likely affect MP uptake, distribution, metabolism, and detoxification processes, ultimately impacting MP abundance in mussel tissues. Furthermore, given that temporal variations have been shown to affect MP spatial distribution in similar species (*Botelho et al., 2023*; *Fraissinet et al., 2024*), incorporating such variations into study protocols is essential. Long-term studies tracking MP accumulation across different life stages and seasons would provide valuable insights.
Since fibers occurring in the environment are not only plastic material, a detailed chemical analysis is necessary to assess the actual microplastic fiber abundance. Therefore, the chemical characterization of MPs is a valuable step after the visual identification. This is particularly important because, as *Rebelein et al. (2021)* reported, on a global scale, two-thirds of the predominantly fibrous items of marine surfaces are MPs, whereas 31% are natural fibers. However, the identification and characterization of MP fibers present several challenges, which ultimately may lead to an underestimation (*Silva et al., 2024*).

Understanding the polymer composition allows researchers to potentially trace the source of the contamination and account for variations in the MP content of sessile animals like mussels from different locations. Raman spectroscopy is a powerful tool for the identification and characterization of microplastics in environmental samples. However, several factors can confound the Raman signal and make it difficult to accurately identify the polymer type (*Silva et al., 2024*). Multicomponent microplastic samples (polymer, fillers, pigments, dyes, *etc.*), UV degradation, and fluorescence are among the most significant challenges (*Lenz et al., 2015*). According to *Joshy, Krupesha Sharma & Mini (2022)*, fibers obtained from the aquatic environment can exhibit fluorescence, absorbance, or band overlay. All of the aforementioned variables can completely mask the polymer spectral peaks making identification practically impossible. Additionally, exposure to environmental factors can significantly alter the chemical composition of polymers (*Teboul et al., 2021*). These factors may have altered the polymer structure of the samples analyzed in this study preventing their chemical identification. Other causes may include the chemical digestion procedures used during sample preparations. Digestion protocols have been extensively discussed to efficiently assess MP abundance and reduce degradation or alteration of the samples (*Dehaut et al., 2016*; *Gago et al., 2018*; *Bessa et al., 2019*) even followed by chemical characterization (*Reguera, Viñas & Gago, 2019*). However, due to the wide variety of MPs nature of environmental samples, no protocol is infallible. Despite the challenges presented by using Raman spectroscopy, it has been shown that this technique presents some advantages over other spectroscopy techniques such as ATR-FTIR (*Cabernard et al., 2018*).

The visual identification criteria used in this study based on the guidelines proposed by *Crawford & Quinn (2017)* suggest that all the analyzed microfibers were MPs, although the majority of the fibers did not display discrete Raman spectra. In our study, Raman micro-spectroscopy confirmed the presence of polyester, polychloroprene, polyacrylonitrile, and polyethylene terephthalate, all synthetic fibers commonly used in the manufacturing of textiles. The importance of combining chemical and visual identification in MP investigations is reinforced by these results. To reduce the uncertainty when no discrete Raman spectra are recorded, future research could include a larger number of samples or include a complementary spectroscopy technique such as FTIR microscopy.

As reported by *United Nations Environment Programme (UNEP) (2018)*, approximately 16% of MPs in the oceans originate from textile laundry. Notably, our study identified pigments like phthalo blue, indigo, orange perinone, and bone black (*Caggiani, Cosentino & Mangone, 2016*; *Infrared and Raman Users Group (IRUG), 2024*) often associated with textile dyeing processes, particularly in denim production (*Zhao et al., 2024*). MPs typically

contain dyes and pigments, and while they can exhibit peaks in the same Raman spectral region as the majority of polymers, their presence can also be a powerful marker of MPs (*Silva et al., 2024*). Hence, microfibers that present signals assigned to the above-mentioned pigments are likely to be also MPs, even though their polymer composition could not be established. The high proportion of blue MPs in S1 may be linked to the nearby wastewater treatment plant, which is a known source of textile-derived microfibers (*Sun et al., 2019*).

## CONCLUSIONS

Our results confirmed the ubiquitous presence of microfibers in mussels throughout the studied marine environment. While it was expected to find a greater abundance of MPs in highly urbanized areas, the study results reveal a more complex profile. The high abundance of microfibers in mussels from S4, a low urbanized area, differs from the typically observed trends of similar studies using bivalves as bioindicators of MP pollution. These results underline the need for more research that follows the protocols and guidelines that have been standardized in recent years, to understand the mechanisms governing MPs transport and accumulation in coastal environments and the factors influencing their spatial distribution. MPs can enter food chains through mussels and can be transferred to higher trophic levels, hence their widespread contamination is concerning. Therefore, further research is required to assess the potential ecological effects of this contaminant on marine ecosystems and human health. Moreover, temporal variability should be considered to evaluate how seasonality related to human activities may influence marine ecosystem contamination and, ultimately, human health.

Ingesting MPs can lead to internal blockages and induce satiation, ultimately reducing the fitness of the mussels (*Wright, Thompson & Galloway, 2013*). *Brachidontes rodriguezii* plays an important ecological role in the intertidal zone, serves as a key food source for various predators, and provides protection and refuge to other species (*Borthagaray & Carranza, 2007*; *Soria et al., 2022*). Thus, the impacts of MPs on *B. rodriguezii* may lead to detrimental effects on other species associated with the mussel beds, which would have an overall negative effect on the ecosystem. A significant extension of the Argentine coastline remains largely unexplored regarding human impact on marine ecosystems. By assessing MP abundance in *B. rodriguezii*, this study aimed to expand our understanding of MP pollution in the Argentine maritime coast, which is crucial for developing effective monitoring and mitigation strategies.

## ACKNOWLEDGEMENTS

The authors would like to thank the anonymous reviewers whose suggestions helped to improve this manuscript.

### Funding

This study was supported by CONICET project PIP 2021-2023 277, ANPCyT project PICT 2021-I-A-726 and UBA project UBACYT 20020220200081BA. The authors received the PeerJ Award at the Argentine Association of Malacology (ASAM) granted at the 4th Argentine Congress of Malacology. The funders had no role in study design, data collection and analysis, decision to publish, or preparation of the manuscript.

### Grant Disclosures

The following grant information was disclosed by the authors:
CONICET: PIP 2021-2023 277.
ANPCyT: PICT 2021-I-A-726.
UBA: UBACYT 20020220200081BA.
PeerJ Award at the Argentine Association of Malacology (ASAM).

### Competing Interests

Daniel Murgida is an Academic Editor for PeerJ.

### Author Contributions

- Lucina Olivia Migliarini performed the experiments, analyzed the data, authored or reviewed drafts of the article, and approved the final draft.
- Sonia Maribel Landro performed the experiments, analyzed the data, prepared figures and/or tables, authored or reviewed drafts of the article, and approved the final draft.
- Freddy Martinez-Espinoza performed the experiments, analyzed the data, authored or reviewed drafts of the article, and approved the final draft.
- Daniel Horacio Murgida performed the experiments, analyzed the data, authored or reviewed drafts of the article, and approved the final draft.
- Florencia Arrighetti conceived and designed the experiments, performed the experiments, analyzed the data, prepared figures and/or tables, authored or reviewed drafts of the article, and approved the final draft.

### Data Availability

The raw measurements are available in the Supplementary Files.

### Supplemental Information

Supplemental information for this article can be found online at http://dx.doi.org/10.7717/peerj.19518#supplemental-information.

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
