# Peer review of "Profiling microplastic fibers in the intertidal sentinel mussel Brachidontes rodriguezii from the coast of Buenos Aires, Argentina"

_PeerJ, doi:10.7717/peerj.19518_

## Round 0.1 · original submission · Minor Revisions

The reviewers have evaluated your manuscript and found that it requires considerable minor revisions. Please provide a detailed point-by-point reply to each of the reviewers' comments.

Reviewer 1 ·

Basic reporting

This manuscript is interesting, especially since it presents data and results about microplastics associated with a bivalve not so studied in regards to this topic. All data seem robust and well-analyzed.
Despite that, I find the text a bit sparse, especially in the introduction. The reader may find it difficult to re-engage with a few significant points if some critical issues are only covered in the discussion and no further context is provided in the introduction. The language structure makes some parts a bit unclear but the manuscript can be well-implemented with some adjustments.
Moreover, different confusing parts need to be rephrased for better comprehension. One of the major critical issues in terms of confusion concerns the continuous alternation in the text of the words microplastics (MP) and microfibers. It becomes difficult to distinguish between microplastics in general, non-fibrous microplastics, and microfibers. The general suggestion would be to find a way to make this clearer especially in some points ( lines: 34-37 / 132-134 / 143 / 156 / 186-192 /200/ 227/ 266 / 278 / 284 /355).

COMMENTS:

Abstract

1. Line 29: The use of the term "efficiently" when referring to the filtration activities of mussels here seems a little bit misleading as if the ecological role of these bivalves is precisely filtering to accumulate microplastics. I would suggest to avoid the use of this word.

2. Lines 34-38: Sentences here are a bit hard to follow. I suggest authors rephrase these lines, i.e.: “The abundance of MPs varied significantly among the stations, with the highest levels observed in mussels from S2 and S4, corresponding to the low-urbanized areas. These findings seem to suggest that factors other than urban pollution, such as agricultural activities and nearby streams, may contribute to MP contamination.” It is also recommended to refrain from using terms like "surprisingly" to maintain a more scientific language.

3. Lines 40-45: This part is a bit confusing. To improve clarity, the suggestion is to rearrange some passages, like: “…identification suggested that all found microfibers were microplastics,…” with something like “…identification suggested that what was observed could be classified as plastic microfibers,…” or “…about 10 % of the analyzed microfibers, revealing the presence of polyester, polychloroprene, polyacrylonitrile, and polyethylene terephthalate.” with something as: “for about 10% of the analyzed microfibers, highlighting the presence of polymers like polyester, polychloroprene, polyacrylonitrile, and polyethylene terephthalate.”

Introduction:

As already highlighted, the work is very interesting but it seems too superficial in some parts, with an introduction a bit sparse. Here a general comment is to try to expand a bit the text by introducing also some of the most important concepts proposed in the discussion (i.e., the fact that synthetic fibers take longer to be egested; that fibers found are not only plastic material; environmental factors that can alter the chemical composition of polymers; the need of a common methodology). Deepening these parts would help the reader to have a more complete picture of the work and to better follow the train of thought from the introduction to the conclusions.

4. Line 51: Please remove the “i.e.” before plastic particles < 5 mm in size.

5. Lines 55-56: Here a suggestion is to slightly broader introduction to microfibers and their properties, emphasizing their importance as one of the most abundant types of microplastics (some references: https://doi.org/10.1177/0040517521991244 / https://doi.org/10.1016/j.cbpc.2021.109196). I would add more information about their origins, composition, and what makes them more "threatening" in some cases than other shapes of microplastics (as reported in lines 241-245), so the connection with results and discussion is easier to make.

6. Line 59: As already reported, avoid the use of the term “efficiently”.

7. Lines 62-63: Please rephrase this sentence for better readability. Furthermore, the suggestion is to expand the concept of these lines to strengthen what the authors propose in this work.

8. Lines 64-66: It would be interesting to briefly explore which methodologies could help most in addressing this critical issue. Moreover, please add the “s” at the end of “present” (line 66).

9. Lines 66-67: To better connect these two paragraphs, the suggestion is to highlight how an efficient methodology for MP detection can be crucial especially if we talk about food security and human health. In these regards finding a way to face the methodological challenges mentioned above can become a true asset.

10. Line 73: Please remove the “s” at the end of “includes”.

Materials & Methods:

11. Line 108: Please remove the comma after “regime”.

12. Lines 110-111: Were chemical, physical, and/or pollution data collected at these stations before collecting the individuals? This would be interesting data especially to understand any correlations with the obtained results.

13. Line 125: Add an “a” between “to” and “sealed”.

14. Line 150: Kindly remove the word “typically” for more formal phrasing.

Results:

15. Line 176: Remove the comma after “mm”.

16. Line 183: Please correct “cero”.

17. Line 185: This line is not very clear. Please, specify more.

18. Lines 199-220: Here there is something quite confusing. Of 239 items classified as microfibers (line186), only 41 were analyzed by Raman spectroscopy. Of these 41 only 4 fibers were confirmed to be so.(?) However, in line 213 is reported that another group of 10 microfibers has been investigated. Are these 10 microfibers included in the 41? Or if considered separately, based on what parameter were they divided? Moreover In lines 223-224, 21 and 6 remaining fibers were analyzed. Are also these included in the 41? It would seem so from calculation, but in line 219 other three fibers are mentioned and in line 222, two cotton microfibers were reported. The way this information is presented makes it difficult to clearly understand what has been done. Therefore, it is suggested to revise the paragraph to better present these results.

Discussion:

19. Line 229: Please correct “it has shown” with “it has been shown”.

20. Line 259: There is a double space between “(2020)” and “found”. Please remove it.

21. Line 225: Please remove the double space between “MP distribution” and the citation.

22. Line 264: Please delete “Unfortunately” for more formal language.

23. Lines 268-272: These are very interesting points regarding this study. The suggestion here is to emphasize a little more the importance of a common methodology to improve future studies and thus allow more comparisons between the results of different research. This may gain further prominence if also presented in the introduction (as suggested in the general comment of the chapter).

24. Line 279: It would be better to remove the word “notably” for more formal phrasing.

25. Line 299: Please remove “Interestingly” to maintain the scientific language.

26. Line 303: In this sentence is reported an interesting hypothesis that deserves more attention. Adding examples and references about it would improve the text.

27. Lines 305-306: I suggest to avoid the repetition of “mussels’ health” in these lines for better readability.

28. Lines 306-310: These lines are very confusing. Is not clear how the reference reported here supports the hypothesis that a higher microfiber abundance causes stress and impairs the mussels’ health. Hypotheses that also involve the absorption of POPs on microplastics emphasizing their effects on mussels are very interesting, but here completely disconnected. Without further insights, examples, and more coherent references, the thread of the discussion is completely missing. Please revise the whole part.

29. Lines 312-313: Here is hypothesized that physiological and biochemical changes during mussels’ life cycle might influence their sensitivity to MPs, but how? Adding examples or references from other studies that have found similar results would make this idea stronger.

30. Line 355: There is a double space between the words “blue” and “MPs”. Please, remove it.

Conclusions:

31. Line 362: Please consider removing “surprising” from the sentence for a more scientific phrasing.

Experimental design

No comment.

Validity of the findings

No comment

Reviewer 2 ·

Basic reporting

- While the writing is generally good, this draft still contain a lot of run-on sentences. This might be a stylistic choice but it greatly hinder the comprehension of the paper. Shorter sentences with no or little modifiers can prove to be more easily understandable to readers.
- References provided were sufficient to support the authors' arguments.
- Geographical location information in Table 1 is repetitive with Figure 1. What would be more useful to readers is the inclusion of general description of the station area (i.e., proximity to ...).

Experimental design

- Research questions were clearly defined and motivation was spelled out.
- The use of ten bivalves from each of the four stations might be too few. Authors also failed to mention how the ten bivalves per site were selected (i.e., random vs systematic sampling).

Validity of the findings

- Despite too few samples, the statistical tests in this study seemed robust.
- The first sentence seems to be an overstatement because this work only studied what was found in one selected species of intertidal bivalves and not other environmental media (i.e., seawater and sediment).

Additional comments

This study investigated microplastic (MP) contamination in mussels along the Mar del Plata coast. Potential microplastics, primarily microfibers, were found in nearly all mussels sampled, with higher concentrations in both urban and a low-urbanized area. Raman spectroscopy confirmed polymer types for only a small percentage of samples, highlighting the limitations of current identification methods. The findings underscore widespread MP pollution and the need for further research to understand sources and impacts, particularly given the observed negative correlation between MP abundance and mussel health. However, this manuscript suffers from several key issues, including limitations in visual and spectroscopic identification, the lack of environmental media sampling, and insufficient consideration of temporal variability.

General Issues:

- Visual Identification:
- Replace "microplastics" with "potential microplastics" when referring to visual observations.
- Acknowledge the need for additional tests (hot needle, fragility) to confirm plastic identity.
- Raman Spectroscopy:
-Address the limitations of Raman spectroscopy and suggest FTIR-microscopy as an alternative or complementary method in the discussion.
- Environmental Media:
- Highlight the lack of MP data from water and sediment, and emphasize how this limits the interpretation of mussel contamination, especially in the low-urbanized area.
- Temporal Variability:
- Discuss the importance of temporal variability and acknowledge the limitations of a single-time study.
- Cite other papers that have shown this variability.
- Writing Style:
- Revise run-on sentences to improve clarity and readability by using shorter sentences.

Line-by-Line Corrections:

Title:
- Remove "Sentinel" for improved clarity.
Introduction:
- Line 73: Replace "benthonic" with "benthic."
- Line 123: Add a brief explanation of the Condition Index (CI).
- Line 133: Clarify the meaning of "equal structure."
- Line 140: Change "Abundance" to "MP abundance."
Methods:
- Line 170: Break the sentence after "...differences among study sites."
- Line 183: correct "cero" to "zero" or whatever the correct word should be.
- Lines 183-185: Divide the first sentence of this paragraph into two sentences for better flow.
Results:
- Line 213: Start a new paragraph here as the previous paragraph is already long enough. Perhaps beginning a new paragraph can help to emphasize the limitations of Raman spectroscopy as shown by your results.
Discussion:
- Line 233: Replace "main type" with a more accurate description, such as "all MPs were fiber-shaped."
- Lines 265-266: Speculate on the reasons for differences compared to Truchet et al. (2021), considering location and species.
- Line 322: Replace "typology" with "types."
Conclusion:
- Revise the first sentence to avoid overstating the scope of the study.
Figures/Table:
- Table 1: Replace repetitive geographical information with general station descriptions (proximity to...).

---

## Round 0.2 · Minor Revisions

Some minor correction need to be addressed. Please provide a point-by-point amendment along with the revised text.

Reviewer 1 ·

Basic reporting

The authors have done a great job in improving the manuscript both in terms of professional English and in the fluency and clarity of the text.

The background of the topics presented has been well covered, providing greater clarity and robustness to the results presented. For example, an interesting background is provided on why standardizing methodologies in this field of study is essential, giving even more relevance to this study.

Experimental design

The objectives and research questions are clear and relevant to the objectives of the journal. The article highlights knowledge gaps to be filled, and the work done introduces new results relevant to the context in which the research was conducted.

Attention has been paid to the accurate description of the methodologies used and the details, making the text a clear source of information, also in terms of replicability in other sites.

Validity of the findings

The manuscript, by filling essential gaps in research on microplastics and their potential impact, contributes to provide new information to support both existing and future literature. The presented analysis of the results is robust and well structured, strengthening the answers to the research questions presented.

Reviewer 2 ·

Basic reporting

Thank you for addressing the reviewers' concerns and comments. Your efforts to revise the manuscript are appreciated. However, some further revisions are still required in the revised version. Please see below.

[Line 32] Add "city" after "...the most popular resort" because Mar del Plata is not a resort but it is a resort city.

[Lines 39-42] Regarding Reviewer#1's comment #3 on the Abstract, the authors indicated they would change the sentence to: "Identification suggested that all found microfibers were plastic, with approximately 10% of the analyzed microfibers revealing the presence of polymers such as polyester, polychloroprene, polyacrylonitrile, and polyethylene terephthalate." However, the revised version still reads: "Visual identification suggested that all found microfibers were plastic microfibers, and approximately 10% of the analyzed microfibers revealed —by Raman spectroscopy— the presence of polymers such as polyester, polychloroprene, polyacrylonitrile, and polyethylene terephthalate." Please ensure this change is implemented in the next revision.

[Lines 56-65]. For better clarity and conciseness, please consider changing the sentences as follows. "These MPs range in size from small to large and occur in various colors (transparent, blue, black, red, etc.). They also display diverse morphotypes, including fibers, pellets, fragments, and films. The toxicity of plastic particles depends on both the quantity and physical characteristics of ingested items, as well as their chemical properties (Leistenschneider et al., 2023). MPs comprise different polymer compositions such as polystyrene (PS), polypropylene (PP), nylon, polyethylene terephthalate (PET), and polyvinyl chloride (PVC). Chemical characterization can be performed using various spectroscopy techniques, including micro-Fourier Transform Infrared spectroscopy (μ-FTIR), Attenuated Total Reflection Fourier Transform Infrared spectroscopy (ATR-FTIR), and micro-Raman spectroscopy (μ-Raman) (Bessa et al., 2019)."

[Lines 67-69] Please change the sentence as follows. "Additionally, their prolonged egestion times in marine organisms raise heightened concern, as this may result in increased exposure and associated risks (Cole et al., 2016)."

[Line 70] Please drop a comma and add an "s" after "organism." So the revised sentence should read: "MP ingestion causes not only physical but also physiological damage to marine organisms (..., ..., ...)."

[Lines 71-76] Please change the sentences as follows. "Due to their filter-feeding mechanism, bivalves accumulate MPs from the water column (Li et al., 2016; Reguera, Viñas & Gago, 2019). Therefore, the detection of MPs in their soft tissues serves as an indicator of the concentration and bioavailability of this pollutant in the environment (Beyer et al., 2017). Furthermore, the accumulation of MPs in bivalves is a human health concern because they can contaminate the human food supply (Li et al., 2016)."

[Line 90] Please delete the comma after "matrices."

[Line 95] Please add "species" after "mussel."

[Line 96] Please add "area" after "resort."

[Lines 142-144] Please change the sentence as follows. "Then, the soft tissues from each individual mussel were added to a separate sealed glass beaker containing 20 mL of 10% KOH per gram of mussel tissue."

[Line 215] Remove an "s" from "microfibers."

[Line 233] To avoid repetition, change "did not correspond to any MP" to "did not match with any polymer."

[Line 247] Add "it" between "that" and "is" to make the phrase read as "...has been shown that it..."

[Line 289] Replace "that" after the comma with "which" to make the sentence grammatically correct.

[Line 292] While the preposition, "within," is grammatically correct in this context, it might not be the most precise or natural choice given the context. I recommend changing it to "across" or "among,"

[Line 315] To make it more clear, change "the unexpected abundance" to "the unexpectedly high abundance" and leave a space between "found" and "in."

[Line 317] Replace "if" with "whether" for clarity.

[Line 336] I am under the impression that you want to use "sensitivity" instead of "sensibility" here. Also, please add a full stop after "...to MPs" to mark the end of the sentence. In addition, leave a space between "MP" and "uptake."

[Line 345] Replace "report" with "reported" to reflect the past tense.

[Line 397] Replace "the last years" with "recent years" to avoid ambiguity.

Experimental design

No further comment.

Validity of the findings

No further comment.

---

## Round 0.3 · accepted · Accept

Thank you for these last final revisions.